# Food insecurity in South Indian households with TB during COVID-19 lockdowns and the impact of nutritional interventions: A qualitative study

**Madeline E. Carwile**[1☉], **Komal Jain**[2☉], **Madolyn R. Dauphinais**[1],
**Prakash Babu Narasimhan**[2], **Kimberly Maloomian**[3,4], **Manju Rajaram**[2], **Chelsie Cintron**[1],
**C. Finn McQuaid**[5], **Lindsey M. Locks**[6,7], **Lora L. Sabin**[7], **Subitha Lakshminarayanan**[2‡*],
**Pranay Sinha**[1,8‡]

1 Boston Medical Center, Boston, Massachusetts, United States of America, 2 Department of Preventive and Social Medicine, Jawaharlal Institute of Postgraduate Medical Education and Research, Puducherry, India, 3 Harvard Medical School (Adjunct), Boston, Massachusetts, United States of America, 4 Kimba's Kitchen, LLC, West Palm Beach, Florida, United States of America, 5 TB Centre and Centre for Mathematical Modelling of Infectious Diseases, Department of Infectious Disease Epidemiology, London School of Hygiene and Tropical Medicine, London, United Kingdom, 6 Department of Health Sciences, Boston University Sargent College of Health and Rehabilitation Sciences, Boston, Massachusetts, United States of America, 7 Department of Global Health, Boston University School of Public Health, Boston, Massachusetts, United States of America, 8 Section of Infectious Diseases, Department of Medicine, Boston University School of Medicine, Boston, Massachusetts, United States of America

☉ Co-first authors.
‡ Co-senior authors.
* subitha.l@gmail.com

## Abstract

Tuberculosis (TB) mortality rose during the COVID-19 pandemic, as did food insecurity worldwide. Undernourished individuals are at increased risk for incident TB disease and TB mortality. This study examines the impact of COVID lockdowns on the food security of households with TB in Southern India. In-depth interviews and focus group discussions were conducted with household contacts enrolled in a nutritional intervention study, conducted in Puducherry and Tamil Nadu (2018–2024), which provided household contacts of persons with TB with a six-month nutritional intervention. During the COVID-19 lockdowns, 78% of households in our study had no income; 67% resorted to distress financing, such as loans, to afford food; and 44% changed their eating habits, such as buying less food or different items. Respondents reported that the provided nutritional intervention improved their food security. Crises such as the COVID-19 pandemic can reduce both food access and diversity, leading to macronutrient and micronutrient deficiencies that increase the risk of TB incidence and mortality. Undernutrition due to food insecurity may have been a driver for hastened disease progression or disease-related morbidity and increased mortality during the pandemic. Nutritional support interventions should be implemented as a part of pandemic response.

**Data availability statement:** Data are available as Supporting information files.

**Funding:** This work was supported by the Warren Alpert Foundation (grant number 6005415) to P. S.; the Burroughs Wellcome Fund/American Society for Tropical Medicine and Hygiene (100000861 and 100001949) postdoctoral fellowship to P. S.; career investment award from the Boston University Chobanian & Avedisian School of Medicine Department of Medicine to P.S.; the National Institute of Allergy and Infectious Diseases (grant number 5T32 AI-052074-13 and K01AI167733-01A1 to P. S.); and the Civilian Research and Development Foundation (grant number DAA3-19-65673-1 and 100004419), as well as federal funds from the government of India's Department of Biotechnology (501100001407) and the Indian Council of Medical Research. The funders had no role in study design, data collection and analysis, decision to publish, or preparation of the manuscript.

**Competing interests:** The authors have declared that no competing interests exist.

## Introduction

Tuberculosis (TB) is the leading infectious killer worldwide [1]. During the COVID-19 pandemic, the reported number of individuals with confirmed TB diagnoses fell, but mortality rose for the first time in two decades. A well-accepted explanation is that increased mortality was due to a reduction in case findings during the height of the pandemic, which resulted in diagnostic and therapeutic delays [2]. While this may be an important factor, it is likely not the sole reason. Nutritional shocks to vulnerable individuals due to the pandemic may have been another important reason for the rise in mortality.

The COVID-19 pandemic led to a substantial increase in food insecurity worldwide [3,4], particularly in high TB burden countries, such as India [5]. According to the SOFI report, between 691 and 783 million people faced food insecurity in 2022, with a midpoint estimate of 735 million. This figure represents an increase of 122 million people since 2019 (before the pandemic began). Food insecurity refers to the lack of consistent access to enough food for an active, healthy life [6]. It is a complex condition that may stem from inconsistent or insufficient food availability, limited resources for acquiring food, and poor utilization of the food available. While food insecurity and undernutrition are not synonymous, food insecurity can lead to inadequate intake of macronutrients and micronutrients, which leads to undernutrition [6]. Undernourished individuals are more likely to progress to active TB disease and experience worse treatment outcomes [7,8].

At the time of the COVID-19 pandemic, our research team was already running a nutritional intervention study in India with a qualitative component. Here we present insights on how COVID-19 affected the food security of households with TB.

## Methods

### Study setting and design

Data were collected and analyzed as a component of the Tuberculosis - Learning about Experience with Nutritional Supplementation (TB LENS) study, a sub-study of the Tuberculosis - Learning the Impact of Nutrition (TB LION) study. TB LION enrolled participants across Puducherry, Cuddalore and Villupuram in southern India, and provided household contacts of persons with TB with a six-month nutritional intervention comprising of food baskets that provided 2600 Kcal/d for all adults and 1300 Kcal/d for children under 13, as well as nutritional counseling [9]. The study population and food insecurity status have been previously described in detail [9]. In brief, approximately 60% of persons with TB in this region are attributable to undernutrition. All household contacts are between 18 and 60 years old, do not have diabetes, and live with a person with microbiologically-confirmed TB.

TB LENS was a qualitative study with a phenomenological approach conducted from June 2021-May 2023, wherein 19 in-depth interviews (IDIs) and two focus group discussions (FGDs) were conducted with household contacts who received the TB LION nutritional intervention. Among the household contacts who participated in the IDIs, six were male and 12 were female, with an average age of 30.9 (SD = 11.7). On average, participating households had 4–5 people. This analysis focuses on responses related to COVID-19 and food security from household contacts. All participants gave their written informed consent for inclusion before they participated in the study. The study was approved by the Institutional Review Boards of Boston University and Boston Medical Center (H-39980) and the JIPMER Internal Ethics Committee.

### Study population

Household contacts were recruited from among TB LION intervention participants using a convenience sampling approach, with a purposive element to ensure gender balance among

participants. Household contacts were approached and enrolled during a TB LION study visit; they were eligible to participate in the study if they had received nutritional interventions as part of the TB LION study. Enrollment was prioritized among those with upcoming visits for efficiency of the study staff and minimized burden for participants. No participants refused to participate or dropped out of the study, and each participant was interviewed once.

## Data collection and analysis

IDIs and FGDs were conducted in Tamil by pairs of Tamil-speaking medical social workers, at least one of whom was the same gender as the participant(s). These medical social workers were trained by senior staff, and had experience conducting qualitative research. They had existing relationships with the study participants as part of the food delivery and health monitoring team, and participants knew minimal professional details about the study staff.

All IDIs and FGDs followed a semi-structured interview approach, allowing for flexibility during the interviews (see S1 Data). IDIs took place in HHCs' homes, in private rooms at nearby clinics, at Jawaharlal Institute for Postgraduate Medical Education and Research (JIPMER), or via phone or video call. FGDs were held at JIPMER and at the Cuddalore District TB Centre. For both IDIs and FGDs, no one else was present other than the researchers and study participants. IDIs were around one hour in duration, while FGDs lasted between one and three hours each. The research plan involved pre-determined numbers of IDIs and FGDs, and saturation was also achieved by the completion of data collection.

Each IDI and FGD was audio-recorded and transcribed in Tamil, then translated into English by bilingual research assistants. Field notes were made during the interviews, and they were referred to at the time of transcript preparation and analysis. English language transcripts were analyzed using NVivo (version 12) [10]. A thematic approach was employed to identify key themes related to study questions iteratively; transcripts were then coded by one analyst (MC), with additional framework generation and coding discussion by another analyst (MRD). Coding was based on common themes and used a hybrid (inductive and deductive) methodology. The transcripts were returned to participants for comments or correction after a week, and the study staff encouraged participants to share feedback on the findings.

## Results

### Background on the COVID lockdowns

In our study region, the initial 21-day lockdown began on March 22, 2020. Residents had limited access to grocery stores; while some received groceries from NGOs or local politicians, many did not. Typically, grocery stores did not open until 10 a.m. at the start of the lockdown, and were only open for half days later in the lockdown. The lockdown measures were strictly enforced by the police. The lockdown was extended through May 31, 2020. Thereafter, restrictions were lifted in a phased manner.

### Reduced income

All participants were affected by the COVID-19 pandemic and its associated lockdown. Of the 19 IDI household contacts, 14 (78%) had no household income during this time, two experienced reduced income, and one had only a single household member working during the lockdown. Of the 13 household contacts whose incomes were recorded, 12 had income prior to the pandemic, and only one was unemployed. One male household contact explained: "there was no income during lockdown period. No income for 6 months. Nothing was running properly. The policemen used to beat [us] if we came out of [our] home. It was the worst."

The impact of lack of income was described by another male household contact, who stated: "we suffered for food because my father and mother had no jobs at that time. We were not able to take nutritious food, but ate just for hunger. We didn't have sufficient money to buy those healthy food items." A female IDI household contact described how "no one went for work from my house. […] We have goats at home. We take care of them and sell them for business. However, because of the lockdown no one purchased the goats. It was very difficult for us."

### Food access, availability, and pricing

Some participants received the intervention before the lockdown period, while others received the intervention after; during lockdown the intervention was halted and re-started later. The intervention was stalled until October 2020 for two participants and December 2020 for one participant. During the lockdowns, participants faced difficulties in accessing food due to limited availability and increased prices, which could have affected their nutritional status by compelling them to eat less food or choose less healthy options.

39% of IDI participants described stores being closed or only open during limited hours, while 33% stated that food prices increased during this time. A male IDI participant stressed the resulting challenge: "we faced a lot of problems getting things from outside. Most of the shops were closed and we had to get [food] from a faraway shop. In those shops also we had to get in after standing in a long queue. Sometimes we did not get enough things because of [having] less money." One female IDI participant stated that stores "used to sell 5 rupees worth of stuff for 7 or 8 [rupees] and blamed it on corona." A male FGD participant described how milk packets normally cost "ten rupees, but no packet was easily available at that time. Not all [were] ten rupees, they used to sell them for fifteen rupees a milk packet." One female household contact in the FGDs stated that "all the vegetables [were] very expensive. That's why I didn't buy any vegetables. I couldn't buy anything like before."

### Coping strategies

Household contact IDI participants described various coping methods to deal with this reduced or lack of income, including taking out loans (67%), changing the type of food eaten (44%), and eating less food (33%). Participants who described changing the type of food eaten referenced eating porridge; no longer purchasing meat; eating only rice; and not being able to afford vegetables. For instance, one female FGD participant stated "we mostly only [ate] porridge."

Those who took out loans often described still having substantial sums to pay off, such as the male IDI participant who stated "during lockdown times, we used my mother's jewelry for loans. We bought food using that money. […] Half of the jewels are [still] in the bank." A female IDI participant described how her family "borrowed a lot […] we are still paying."

### Support provided by nutritional intervention

Both IDI and FGD participants described the health and financial benefits of the nutritional intervention. For instance, one female IDI participant stated that "after [eating] the given nutritious items, we felt better and we improved so much. […] We gained so much weight." A male IDI participant stated "when compared to before we all are healthy now. My brother used to be skinny; now, he looks like he has gained weight." Participants also connected the intervention to improved food security during the COVID pandemic. As a female IDI participant explained: "at that time you gave us the food supplies, it was useful. We did not face much difficulty because of that. No one at home was employed at that point. We cooked and ate from the food supplies you gave us."

## Discussion

By collecting qualitative data as part of an ongoing trial, our study was able to gain insights into the hunger-related impacts of the COVID lockdown in a highly vulnerable population in India. Our study revealed that 78% of households in our study had no income, 67% resorted to distress financing to afford food, and 44% changed their eating habits during the COVID-19 pandemic-related lockdowns. Further, there was a clear reduction in dietary diversity with decreased intake of protein and several micronutrients. Based on these data, we believe the COVID-19 related food insecurity may have led to increased TB mortality in two distinct ways: first, the increase in undernutrition may have caused higher incident TB in household contacts, thereby increasing mortality due to the higher overall number of people with TB. Second, increased undernutrition may have increased mortality in persons who already had TB disease.

Undernourished individuals have impaired adaptive and immune responses, a condition that has been described as nutritionally acquired immune deficiency (N-AIDS) [11]. Most previous studies linking undernutrition and TB have largely used body mass index (BMI) as a measure of undernutrition. A dose-dependent relationship between BMI and TB disease incidence has been established. One systematic review proposed a log-linear relationship wherein $1\,kg/m^2$ decrease in BMI can result in a 14% increased risk of TB disease incidence between the BMI range of $18–30\,kg/m^2$ [12]. This relationship may become non-linear at lower BMI ranges [13]. Food insecurity over months can result in acute weight loss and increase the risk of incident TB disease.

An individual's BMI provides only a limited understanding of their nutritional status and correlates poorly with micronutrient deficiencies. We found that during periods of nutritional shock, individuals often substitute meat and vegetables with cereals. This can result in preserved or even increased BMIs alongside underlying micronutrient deficiencies. A Peruvian cohort showed that deficiencies in vitamin A and vitamin E were associated with a 10-fold and 3-fold increased risk of incident TB disease respectively, after adjusting for numerous risk factors including BMI [14,15]. Other micronutrients such as vitamin D, B12, zinc, and selenium also play an important role in the immune response and may be diminished in the setting of pandemic lockdowns [16]. Decline in dietary diversity may have thus increased the risk of incident TB disease even in individuals who did not lose weight. Notably, given the profound economic shock of the lockdown, reduced dietary diversity may have persisted beyond the period of lockdown, further increasing TB risk.

In addition to increasing the risk of incident TB disease, undernutrition has also been linked to unfavorable TB treatment outcomes such as death, treatment failure, and relapse [7]. The risk is particularly high for individuals with BMI $<17\,kg/m^2$. These outcomes are likely driven by increased risk of cavitation and lung involvement, increased risk of drug toxicity, and poor engagement with care due to hunger [11]. Acute nutritional shocks, like the one observed in the Dutch hunger winter, a severe food shortage caused by Nazi blockades of Dutch cities during World War II, have the potential to impact TB mortality rates in affected populations over months [17]. The months-long food insecurity faced by persons with TB during the COVID-19 pandemic may have increased their risk of mortality. Vulnerable households with TB may have faced comparable conditions during pandemic lockdowns.

Fortunately, these risks can be mitigated in large part through food security interventions. While the TB LION is focused on exploring the immunological impacts of nutritional support, larger studies in India have established a causal link between improved food access and reductions in TB disease incidence by providing macronutrient and micronutrient support through food baskets [18]. Indeed, the RATIONS trial demonstrated that providing food baskets reduced TB disease incidence by approximately 40% among household contacts of

persons with TB disease. Persons with TB in the RATIONS trial also had higher weight gain and possibly lower mortality compared to previous cohorts [7,19]. More recently, a cluster-randomized study by Mahapatra et al. in Odisha, India demonstrated that nutritional support improved treatment success by approximately 40% [20]. The INSTITUT study in Benin and Togo is currently studying the impact of nutritional support on treatment outcomes and post-TB lung disease [21].

Distributing nutritious and locally acceptable in-kind support during future lockdowns is likely to help mitigate TB disease and may also have myriad medical and non-medical benefits by lessening the economic and nutritional shock of the pandemic. Indeed, our study participants reported selling property and taking loans to buy food. Loans from traditional money lenders can have usurious rates that can have devastatingeconomic consequences.

The neighboring state of Kerala provides a case study of mitigating the nutritional shock of the COVID-19 lockdowns by providing food baskets containing dry rations. These baskets proved to be a feasible and well-accepted intervention that helped reduce food insecurity and allowed individuals to shelter safely during the lockdown [22]. In addition to food grain, the Kerala food kit also contained condiments, spices, and tea to ensure good utilization.

Our study has several limitations. The opinions expressed are those of a small group of individuals. This study focused on TB-affected households in Tamil Nadu and Pondicherry and may not be representative of households in other areas of India. Additionally, participants may have been reticent to criticize the nutritional intervention received through our interventional study because the social workers conducting the interview were also part of the TB LION study. The IDIs and FGDs being led by study staff may have introduced biased responses; however, due to the design of the parent study, performing them within the given timeframe would not have otherwise been possible. Reflexivity may have been a concern, both on the part of the interviewers and on the part of the analysts. This was minimized by collaborating as a team on interview guides and codebooks, and using an adapted acceptability framework to maintain internal consistency.

We suggest that future studies be conducted with larger and more diverse populations to understand the experience and needs of households with TB and other vulnerable households during pandemic lockdowns, as well as other economic and humanitarian crises that affect food security.

While factors such as late detection and notification of cases and other health system disruptions were major contributors to the increase in TB mortality, undernutrition is an overlooked modifiable factor. Undernutrition, food insecurity, and poverty caused by the pandemic could continue to increase TB morbidity and mortality in coming years, even as COVID-19 cases slow [23]. Our data provide compelling evidence that social protections in the form of in-kind nutritional support should be a standard component of pandemic response plans.

## Supporting information

**S1 Data.  Transcripts of the COVID-19-related material from the focus group discussions and in-depth interviews.** Material not related to the focus of this manuscript has been removed.
(DOCX)

**S1 File.  In-depth interview questions for household contacts.** The COVID-19-related questions have been highlighted in yellow.
(DOCX)

**S2 File. Focus group discussion guide.** The COVID-19-related questions have been highlighted in yellow.
(DOCX)

**S1 Checklist. Inclusivity in global research.**
(DOCX)

## Author contributions

**Conceptualization:** Madeline E. Carwile, Prakash Babu Narasimhan, Kimberly Maloomian, Chelsie Cintron, Lindsey M. Locks, Lora L. Sabin, Subitha Lakshminarayanan, Pranay Sinha.

**Data curation:** Madeline E. Carwile, Komal Jain, Manju Rajaram.

**Formal analysis:** Madeline E. Carwile, Pranay Sinha.

**Funding acquisition:** Pranay Sinha.

**Investigation:** Madeline E. Carwile, Komal Jain, Madolyn R. Dauphinais, Manju Rajaram, Subitha Lakshminarayanan, Pranay Sinha.

**Methodology:** Madeline E. Carwile, Komal Jain, Manju Rajaram, Subitha Lakshminarayanan, Pranay Sinha.

**Project administration:** Madeline E. Carwile, Komal Jain, Madolyn R. Dauphinais, Subitha Lakshminarayanan, Pranay Sinha.

**Software:** Madeline E. Carwile.

**Supervision:** Subitha Lakshminarayanan, Pranay Sinha.

**Writing – original draft:** Madeline E. Carwile, Pranay Sinha.

**Writing – review & editing:** Madeline E. Carwile, Madolyn R. Dauphinais, Kimberly Maloomian, C. Finn McQuaid, Lindsey M. Locks, Lora L. Sabin, Subitha Lakshminarayanan, Pranay Sinha.

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
