## [Decision Letter · Decision Letter 0]

10 Sep 2024

PGPH-D-24-01272

The COVID-19 pandemic lockdown exacerbated food insecurity among households with TB in South India: possible implications for increased TB mortality

Dear Dr. Sinha,

Thank you for submitting your manuscript to PLOS Global Public Health. After careful consideration, we feel that it has merit but does not fully meet PLOS Global Public Health’s publication criteria as it currently stands. Therefore, we invite you to submit a revised version of the manuscript that addresses the points raised during the review process.

We look forward to receiving your revised manuscript.

Kind regards,

Prashanth Nuggehalli Srinivas, MBBS, MPH, PhD

Academic Editor

Journal Requirements:

2. In the online submission form, you indicated that "Data are available upon request.". 

a. In a public repository, 

b. Within the manuscript itself, or 

c. Uploaded as supplementary information.

Additional Editor Comments (if provided):

Reviewers' comments:

Reviewer's Responses to Questions

**Comments to the Author**

1. Does this manuscript meet PLOS Global Public Health’s publication criteria ? Is the manuscript technically sound, and do the data support the conclusions? The manuscript must describe methodologically and ethically rigorous research with conclusions that are appropriately drawn based on the data presented.

Reviewer #1: Partly

Reviewer #2: Yes

2. Has the statistical analysis been performed appropriately and rigorously?

Reviewer #1: I don't know

Reviewer #2: N/A

3. Have the authors made all data underlying the findings in their manuscript fully available (please refer to the Data Availability Statement at the start of the manuscript PDF file)?

Reviewer #1: No

Reviewer #2: Yes

4. Is the manuscript presented in an intelligible fashion and written in standard English?

Reviewer #1: Yes

Reviewer #2: Yes

5. Review Comments to the Author

Reviewer #1: At the outset, I must congratulate the authors and the research team for the work on TB - a very important public health hazard which is one of the leading causes of death all around the world, more so in India. The underlying causes like undernutrition and its impact are well studied and your study about food insecurity during the time of the pandemic definitely worth of notice.

General Suggestions:

1.In the earlier sections of the abstract, there is mention of TB LION and LENS without being referenced or paraphrased in the description which makes it difficult to follow. Is TB LENS the current study in question?

2. I feel that the readers would need a description of what the pandemic lockdown was, how long it lasted, what were the existing sanctions for collection of food, water etc and the said events during your study.

3. We have been only provided with the "LENS questionnaire" used in the available data set for supporting documents which I don't think are completely relevant with the results you have presented as they cater only to two questions mentioned in the C part of the document for IDI. No mention of FGDs and how they were conducted!.

4. The full and the short title title would need an edit and upgrade as its not very reflective of the study in question and there is no mention of the nutritional intervention and its impact as well during the pandemic as you have only selected household contacts of patients with TB if they have received the nutritional intervention!

Specific Suggestions:

37-38: would suggest change to " driver for hastened disease progression or disease related morbidity and increased mortality during the "

47-48: Since the intervention is not part of the study being described, featuring it as part of the results of the study seems and leads to a lot of grey area.

87: Referencing the study population to another study which describes the patient with TB and their household contacts may not be sufficient.

97 : Study population will need some clarity on the demographics of the household contacts you have selected, the age groups, number of the people in the household, as it is difficult to understand the results without having base information about the same.

118-119: Relevance of the policemen statement without context?

129 - Duration of the period during which the intervention was stalled must be written, as it a significant confounder!

Looking forward to reading the revised manuscript with the supporting documents.All the best!

Reviewer #2: This study examines other factors that contributed to the increase in TB mortality reported during the COVID-19 pandemic beyond late TB detection and notification of cases that resulted from disrupted health systems. The authors argued that undernutrition due to food insecurity because of the COVID lockdowns in South India could be an overlooked modifiable factor for increased mortality during this period. They expressed their view in a free flowing and lucid manner that builds on existing body of knowledge from previous studies. These include food insecurity resulting from the COVID-19 pandemic, and effects of undernutrition during periods of food insecurity in increasing TB incidence (due to nutritionally acquired immune deficiency) and unfavorable TB treatment outcomes, including mortality from cavitation and lung involvement, increased risk of drug toxicity, and poor engagement with care due to hunger. The authors highlighted key limitations to the study including small number of participants and response bias that could arise from the participants being inhibited in their responses, since they were beneficiaries of a nutritional intervention program supported by the interviewers.

The authors could do a better job of aligning the purpose and result of the study to the tittle. Whereas the title refers to the implications of food insecurity on increased TB mortality in South India, the purpose of the study, as described in the abstract section, points to the impact of COVID lockdowns on food security. In a similar vein, the results section highlights the reduced income, attendant food insecurity and coping strategies, as well as the benefits of nutritional interventions to the participants. There was no supporting data from the study region or participants that explicitly indicated lesser TB mortality rates among beneficiaries of the nutritional interventions. At best, the inference between food insecurity and increased mortality, is based on other studies that indicated reduced TB incidence from improved macro and micro nutrition (the RATIONS trial) and unfavorable treatment outcome from undernutrition (the LION study). The authors should consider adding supportive data to indicate lower TB mortality among the study population that benefitted from the nutritional supplementation.

6. PLOS authors have the option to publish the peer review history of their article (what does this mean? ). If published, this will include your full peer review and any attached files.

**Do you want your identity to be public for this peer review?** For information about this choice, including consent withdrawal, please see our Privacy Policy .

Reviewer #1: **Yes: ** Nikith Austin DSouza

Reviewer #2: **Yes: ** Eneogu Rupert Amanze

---

## [Decision Letter · Decision Letter 1]

11 Dec 2024

PGPH-D-24-01272R1

Food insecurity in South Indian households with TB during COVID-19 lockdowns and the impact of nutritional interventions: a qualitative study

Dear Dr. Sinha,

Thank you for submitting your manuscript to PLOS Global Public Health. After careful consideration, we feel that it has merit but does not fully meet PLOS Global Public Health’s publication criteria as it currently stands. Therefore, we invite you to submit a revised version of the manuscript that addresses the points raised during the review process.

Thank you for responding to the previous reviewer comments. The manuscript is improved, however, we do note that the Methods section is a little thin. Whilst I appreciate you referred to the TB LION publication in your Methods, since this publication does not describe any qualitative methods. It would be beneficial to include details of any relevant considerations for the recruitment of participants into the study reported here, and include some details on reflexivity, discussion of data saturation etc. To accompany your revision, please complete and upload a copy of the COREQ checklist.

We look forward to receiving your revised manuscript.

Kind regards,

Marianne Clemence

Staff Editor

Journal Requirements:

Additional Editor Comments (if provided):

Reviewers' comments:

Reviewer's Responses to Questions

**Comments to the Author**

1. If the authors have adequately addressed your comments raised in a previous round of review and you feel that this manuscript is now acceptable for publication, you may indicate that here to bypass the “Comments to the Author” section, enter your conflict of interest statement in the “Confidential to Editor” section, and submit your "Accept" recommendation.

Reviewer #1: All comments have been addressed

2. Does this manuscript meet PLOS Global Public Health’s publication criteria ? Is the manuscript technically sound, and do the data support the conclusions? The manuscript must describe methodologically and ethically rigorous research with conclusions that are appropriately drawn based on the data presented.

Reviewer #1: Yes

3. Has the statistical analysis been performed appropriately and rigorously?

Reviewer #1: Yes

4. Have the authors made all data underlying the findings in their manuscript fully available (please refer to the Data Availability Statement at the start of the manuscript PDF file)?

Reviewer #1: Yes

5. Is the manuscript presented in an intelligible fashion and written in standard English?

Reviewer #1: Yes

6. Review Comments to the Author

Reviewer #1: Thank you for working on the edits and recommendations provided.

A well revised manuscript!

All queries are well answered.

7. PLOS authors have the option to publish the peer review history of their article (what does this mean? ). If published, this will include your full peer review and any attached files.

**Do you want your identity to be public for this peer review?** For information about this choice, including consent withdrawal, please see our Privacy Policy .

Reviewer #1: **Yes: ** Nikith Austin D'Souza

---

## [Editor Report · Decision Letter 2]

14 Jan 2025

Food insecurity in South Indian households with TB during COVID-19 lockdowns and the impact of nutritional interventions: a qualitative study

PGPH-D-24-01272R2

Dear Dr. Sinha,

We are pleased to inform you that your manuscript 'Food insecurity in South Indian households with TB during COVID-19 lockdowns and the impact of nutritional interventions: a qualitative study' has been provisionally accepted for publication in PLOS Global Public Health.

Best regards,

Julia Robinson

Executive Editor